# Parametric Integration of Multiple Criteria from a Cultural Heritage Perspective

Suzanne Segeur-Villanueva [1],*, Natalia Caicedo-Llano [1], Roberta Zarcone [2], Aly Abdelmagid [2] and Nicolas Sabogal-Guachetá [3],*

[1] Departamento de Planificación y Ordenamiento Territorial, Facultad de Ciencias de la Construcción y Ordenamiento Territorial, Universidad Tecnológica Metropolitana, Dieciocho 161, Santiago 8330383, Chile; ncaicedo@utem.cl

[2] Laboratoire Géométrie-Structure-Architecture, Ecole Nationale Supérieure d'Architecture Paris-Malaquais, 14 Rue Bonaparte, 75006 Paris, France; roberta.zarcone@paris-malaquais.archi.fr (R.Z.); aly.abdelmagid@paris-malaquais.archi.fr (A.A.)

[3] School of Architecture, University of Notre Dame, Notre Dame, IN 46556, USA

* Correspondence: suzanne.segeur@utem.cl (S.S.-V.); arqsabogal@gmail.com (N.S.-G.); Tel.: +56-98-149-4580 (S.S.-V.)

**Abstract:** Parametric design is a versatile decision-making methodology that allows multi-criteria optimization. However, it is not as common for addressing aspects such as the cultural heritage of a given community. In this context, qualitative research proposes linking a cultural heritage classification with parametric design algorithms that include a stage of "design thinking" methodology, which culminates in community validation. This paper aims to demonstrate the potential of parametric and low-tech design as a multi-criteria decision strategy. Algorithms were developed with the mechanical properties of a given material, with meteorological information as input data, geometry as response, and with a simultaneously integrated multi-criteria decision-making process to improve the design. Most algorithms take as input variables such as size, function, or geometry and, as output, the structural material that best fits them, but not the opposite. This methodology was tested on a case study with the Misak community in Colombia, using Guadua (*Guadua angustifolia*), a subfamily of the bamboo, but which is little used. These criteria provided multiple design alternatives that were constantly debated to adjust and test the parameters of the prototype. The principal outcome is that an existing cultural heritage classification allows for the parametric model's generalization ability.

**Keywords:** technology; ancestral techniques; simulations; design methodology

## 1. Introduction

"Technology is the answer. But what was the question?" When British architect Cedric Price provocatively opened a conference in Oxford in 1966, he was questioning the place of technology in the design and production of architecture in the second half of the twentieth century [1]. This question is even more common today if we consider the extent of the impact of information technologies on social, economic, and industrial structures.

Architecture, as a discipline at the frontiers of science, art, and technology, is at the heart of these issues. It deals with people and communities, responding to their needs and interacting with the environment. If, on the one hand, information technologies have transformed architecture by improving the precision and efficiency of the design process through the integration of information throughout all stages of the project process, facilitating collaboration and offering new manufacturing possibilities, on the other hand, architecture, with its privileged position, has the potential to partner this process of change, to assess its impact and even to predict it by driving innovation [2].

The digital revolution has broadened the field of knowledge by revitalizing the complex relationship between design and construction, giving a central role to the constructive dimension of the architectural project. A set of variables that influence decision making are

produced by the information flow that each of these dimensions creates. Through creation, assessment, ordering, and choosing, these performance-based strategies enable the examination of architectural alternatives utilizing computer-aided design methodologies [3].

Energy efficiency, design, structural, and material factors increase the number of parameters that need to be analyzed during the development of a design, making decision making more complex. The use of algorithm-based modelling takes these challenges and addresses them in parametric design and its principles: "generative mechanism, design constraint, and rule-based design ", creating an ecosystem where they merge to deliver multiple solutions [4].

The parametric design enables designers to manage complicated project information with ease, and modern technologies provide a way to increase productivity in the design and delivery process. In addition, new design technology provides chances for creativity by enabling the designer to increase the range of constructible forms through new design and production techniques [5]. Nevertheless, parametric design is not suitable in every case; sometimes it results in a formalism flooding tendency [6] or sometimes aspects are more difficult to transform into algorithmic rules, such as the culture of a given community.

In parametric design, algorithms and rules expressing "divergent" or "convergent" processes represent parameters [7]. According to Suyoto [8], when design is "divergent", it is based on parameters that develop a design framework associated with creativity. When design is "convergent", it uses a set of principles that converge in a project based on a previously existing design. In both, the decision-making process takes place in four stages: changing parameters, perceiving the geometry, introducing algebraic ideas, and evaluating the geometry. These stages take place in iterative cycles of generation and evaluation, to arrive at the final design solution, where decision making is a process in the development of the project [7].

This method aims to build the parametric model from scratch with a multi-criteria perspective and then re-evaluate it through a performance simulation. Based on the simulation results, the optimizer creates a new design solution and redefines the parametric model. The literature reveals many methods with reference to hierarchical relationships between variables. With each variable, there are many optimization cycles in the first one. In the alternative, designers combine the variables into an integer parameter [9]. An example of such design processes with successive iterations between generation and evaluation is discussed in detail in the research of Tabadkani [10], where the design process of an adaptive solar façade (ASF) using multi-factor parametric design is studied, focusing on geometric form with respect to visual comfort. The design process considered four stages: parameter-based modelling, performance evaluation, simulations, and a final optimization in which data from 1800 options were cross-checked with climatic information and visual comfort, within the design objectives.

While it is frequently used to incorporate climatic, geographical, and ecological criteria, as well as to address form, structure, and material issues, parametric design is less frequently used to create projects that reflect the traditions, culture, and idiosyncrasies of communities. Examples of this include the ICD/ITKE kiosks at the University of Stuttgart in 2010, and the Abu Dhabi Louvre Museum, based on a deep study of the construction of the reed method in southern Iraq and a reinterpretation of the traditional pattern of Arab architecture, respectively [6]. In all of them, the team previously selected identity criteria that were then transformed into algorithms, like geometrical identity forms or traditional materials and constructive systems. This methodology is applied in stages: (1) empathize, based on interviews and research; (2) define objectives, roles, and people; (3) ideate, share ideas, prioritize, and seek "divergence" and "convergence"; (4) prototype, create models with rapid iterations, low quality, or initial designs; and (5) test and evaluate its performance with fast iterations. The methodological model is non-linear, and allows for returning to previous stages once they have been carried out, which allows multiple modelling prior to final prototype, as shown in Figure 1 [11–13].

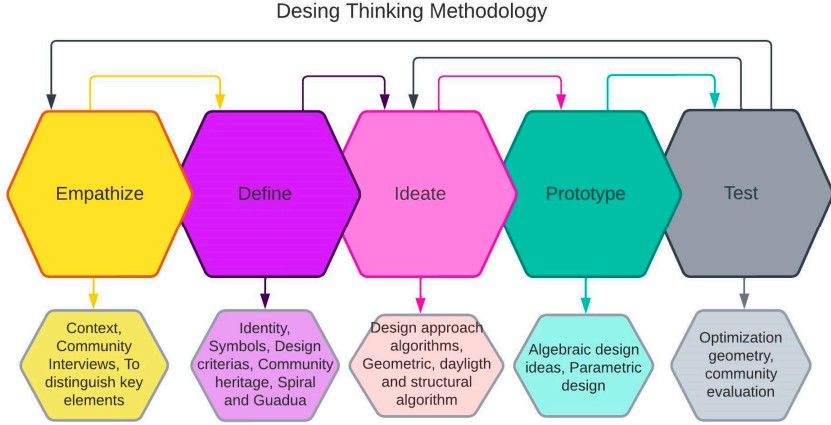

**Figure 1.** General methodology. Own elaboration.

The pillars of "design thinking" place the user as a fundamental element in creating solutions from design or enhancing "user-centered design" [13–15].

When using this methodology with parametric design, a new work process is developed. It consists of a first stage: (1) empathize, where the problem and possible solutions are studied with the users; (2) define, in which the change of parameters that will shape the geometry of the initial design are developed; (3) design, in which the initial proposals are created and a perception of the methodology to be used is carried out; (4) prototyping, in which the parameters are transformed into algebraic design ideas for evaluation in a final stage; and (5) testing, in which the prototype is presented to the community, and design optimization and geometry evaluation processes are carried out.

According to Kamelnia and Hanachi [16] cultural heritage in digital design should be divided into four categories: (1) "Cultural, historical, vernacular"; (2) "Form and symbol"; (3) "Architectural Signatures"; and (4) "Material, Colors".

This theoretical framework and the state of the art help us to develop a methodology of parametric design applied in a case study in Colombia that we are going to describe in the following part.

Colombia is a region home to a wide range of flora and fauna, which supports several distinct ecosystems. The forest near the Ecuadorian line benefits from a tropical climate with consistent daylight, with some dry and wet areas. The Andes Mountain Range's wide range of altitudes support diverse climatic conditions, resulting in various ecological hotspots.

We decided to work in Silvia -Cauca-, a small village of the Misak, an indigenous community that is ecologically, socially, and culturally vulnerable. Numerous factors have affected it, such as globalization, industrial extractivism, and violence linked to drug trafficking. However, this region became a more peaceful place to live again after the signing of the "peace agreement" in 2016 between the Colombian government and the FARC-EP guerrilla group. The Misak community is very well organized and has put a lot of effort into preserving its heritage. This community is located in the region known as the "Eje Cafetero", where there are approximately 70,000 acres of Guadua (*Guadua angustifolia*), but only 30% of this material is used locally [17]. Unlike trees, bamboos in general, and Guadua (*Guadua angustifolia*) in particular, have a rapid growth rate and high productivity. The growth cycle is usually one-third that of a typical tree, and is noted for its rapidity. Guadua (*Guadua angustifolia*) stands out among the top 20 bamboo species in the world compared to American bamboo because of its superior mechanical properties and large size. This species can reach a maximum height of 25 to 30 m in a single year [18], after which its branches and leaves continue to grow for two to three years before it is ready for construction. The Guadua (*Guadua angustifolia*), is a cellulose linear material, with very little natural resistance to degradation. Untreated Guadua (*Guadua angustifolia*) structures are viewed as temporary, with an expected life of no more than five years. Its life can be

increased with preservation treatments. There are traditional methods that do not present significant long- or medium-term benefits; however, chemical methods have better results. Within the most well known are: (1) internodal injection of creosote oil, (2) dip diffusion with boric acid and borax, (3) the hot and cold creosote method, and (4) the Boucherie method, using boric acid and borax [19]. One of the parameters of greatest impact on the durability is its moisture content, which must be below 20% [20]. In addition, it has a strong cortex formed by epidermal cells that are covered by a layer of wax, and this layer naturally protects it from physical damage, prevents the penetration in the lateral form of any liquid [21], and prevents evaporation of the water.

The hypothesis of this research is that linking the "design thinking" methodology to the stages of parametric design allows us to include, among the multi-criteria optimization, the experience, and traditions of the community as a collaborative work strategy. It is tested through the design of a Guadua (*Guadua angustifolia*) pavilion for the Misak Community in Colombia. It also supports the notion that this research is based on, and that the physical and digital processes are intertwined with; architecture is not the sole output of design, but is informed by its relationship with the conceptual and material structure and the production process. Architectural design is enriched by information obtained from its physical context, as well as from collaborative strategies with the community.

In this context, it was examined how the collaborative work methodology allows cultural criteria to be integrated into the digital design process, by studying aspects such as the relationship between form, strength, and material. Also, the study of the order of variables is integrated into the equation to define whether the material is an element that adapts to the form or whether it is is the relationship between form, strength, and material. Digital technology is the tool that allows us to take advantage of it, by utilizing additional resources and creating innovative, tailored solutions that better address actual needs while considering the various stakeholder concerns.

The methodology we propose is based on data-driven design, which allows working on interoperability through the exchange of information. In this article, we present an architectural design methodology capable of integrating information collaboratively with the community from the design phase to improve the relationship between physical and digital processes and to test the control of information flow in relation to its conceptual, material, and constructive framework. The approach uses geometric modelling as the basis for knowledge and decisions; it enables a continuous flow of information from idea to fabrication, integrating the search for the optimal solution into the set of acceptable solutions, and obtaining feedback on ideas through testing on digital models and focus groups within the community [22].

## 2. Materials and Methods

We will outline the methodology creation process based on the "design thinking" methodology:

(1) Empathize: through several activities of space and territory recognition carried out by researchers and the Misak community, providing explanations of the cultural heritage linked to space, symbolism, and lifestyle, so they can comprehend their cosmovision and highlight three key elements.

(2) Define: the starting point is the demands of the Misak members. The first key is the spiral, which represents how a baby's life spins from the moment of his birth until the day of his death. Evidence of this is that the Misak members don a traditional hat with a spiral-shaped knitting detail, shown in Figure 2—described in detail in Section 3.1.1. The second key is the presence of the fire that serves as a metaphor for how people and their possessions interact. The third key component is the importance of natural materials in their architecture. Besides these previous key components, other characteristics need to be considered. The location of the site was chosen because it is at the summit of a mountain, putting people on a site that could be easily recognized from far away, and transforming its meaning into a building of

reference for the community. The location also offers a beautiful view of a breathtaking landscape. The structure serves as a gathering place for ADA members of the Misak Community, including blind or deaf or those who are unable to walk.

(3) Design: the geometric and mechanical characteristics of the Guadua (*Guadua angustifolia*) serve as the primary inputs for the algorithms developed for this research. Thus, we find a geometry using a specific material instead of using materials that are recommended for a group of geometries.

(4) Prototyping: in which the parameters are transformed into algebraic design ideas. The tool used for parametric design is the graphical algorithm editor Grasshopper, which is tightly integrated with 3-DRhinoceros. The first consideration is the maximum slope permitted by the Colombian Code, which is 10% [23]. The second consideration is the space which must have a fireplace that can be placed inside it, with respect to integrating the cosmovision of the Misak Community.

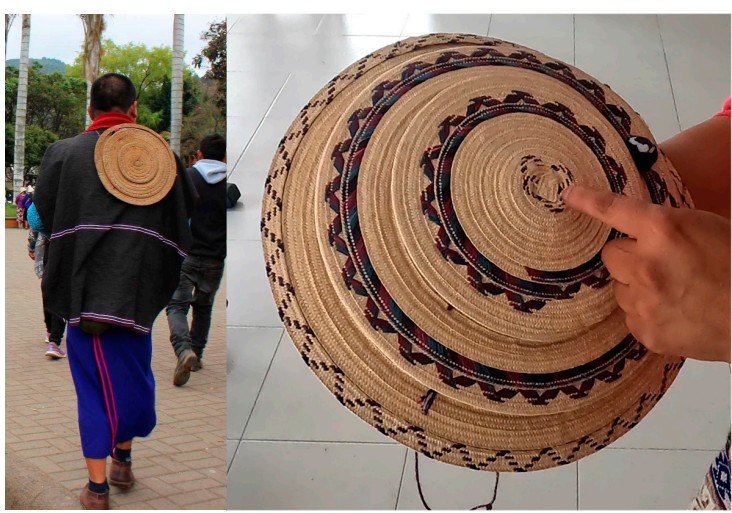

**Figure 2.** Traditional cloth of the Misak Community.

During the form-finding phase of the roof morphology, we considered the design of two variants of a double-curvature surface. In the first variant, called "Variant A" from now on, freeform surfaces were contemplated for the possibility of deforming the rods within the elastic limit. In the second variant, called from now on "Variant B", there are constant-slope surfaces, which do not require the Guadua (*Guadua angustifolia*) to be bowed, considering that only linear structural elements can be used. Once the morphological variants were defined and the material data were integrated into the algorithm, the next step on the design stage transformed the geometry into a structure. This associates architectural intentions—in relation, for example, to environmental parameters such as luminous flux—with structural constraints and the feasibility and ease of construction on site. The final structure must follow the usual dimensioning criteria for a structure meant to be open to the public: the strength and rigidity of the structure must be those of a building considered safe by the standards in force in Colombia. To check that displacements remain under control and stresses below a certain threshold, we use RDM7, under a combination of normal loads—the serviceability limit state—and low probability loads (the ultimate limit state).

(5) Testing: the stage where the prototype design, optimization and geometry evaluation processes are carried out. Information used as input for the daylight algorithm is related to the site location. It is situated at a height of 2.800 m (9.186 foot) above sea level in Silvia, Colombia, which experiences a tropical climate. The average temperature in Silvia is 19 °C (66.2 °F) all year long, and the average wind speed is 13 km/h (8 miles/h) from the north. Since there are no seasons in the tropical climate, a simulation of temperature would be useless. Accordingly, daylight simulations

are essential for decision making about the space between roofs, the separation between walls and columns, the number of walls required, and the number of columns and windows. The daylight algorithm was created using Ladybug, a third-party component of Grasshopper. Finally, the prototype is presented to the community for further iterations. The "design thinking" methodology is shown in Figure 1.

*Method Used for the Interaction between Algorithms*

The following is a description of the interactions between our three different algorithms. It enables us to switch back and forth between decisions in three different areas: geometry (in purple), material (in green) and daylight (in pink). The geometry algorithm provides geometric results, whereas the other two algorithms provide numerical results, as shown in this Figure. When it comes to algorithms that have a numerical result, the daylight algorithm's outputs become inputs, and vice versa, as shown in Figure 3.

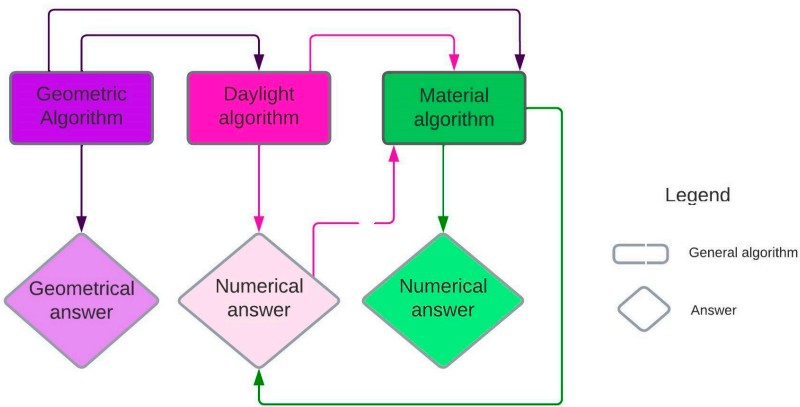

**Figure 3.** Algorithms interactions. Own elaboration.

A specific diagram of interaction within a geometric answer algorithm is shown in Figure 4, using the same color code that has been used in the previous Figure, where geometric elements are in purple. We start by creating a structure whose initial geometry is a spiral. Due to the steep slope of the chosen site, the algorithm incorporates a requirement for ADA-compliant levels inside the building. A fireplace must be included in the algorithm, as well. There are a huge variety of possible geometries when all those elements are combined. A team of experts has examined most of these options and selected the best one.

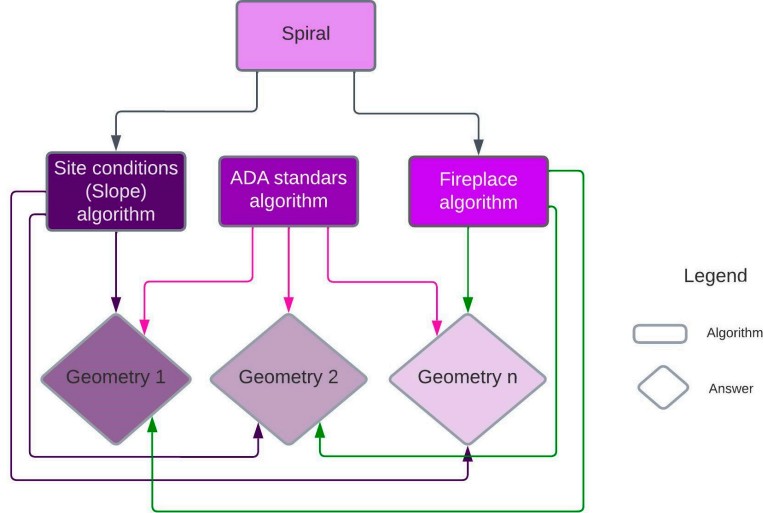

**Figure 4.** Specific diagram of interaction inside a geometric answer algorithm. Own elaboration.

A specific diagram of the interactions between algorithms for numerical answers is shown in Figure 5. A material algorithm (in green) and a daylight algorithm (in pink) are the two main algorithms. These formulas produce a geometric solution that can be expressed numerically. As an output, this number can be used as an input by the next algorithm. Specific algorithms (in purple) show the areas where the two algorithms interact. When the roof does not meet the minimal radius of curvature, some notches in the Guadua (*Guadua angustofolia*) are needed. For the purposes of this research, all earlier calculations were made before the material algorithm was completed. However, we chose not to implement the algorithm because the Misak community lacks the skilled labor and equipment that would be required to fabricate this kind of structure. Dotted lines are used to depict this portion of the work.

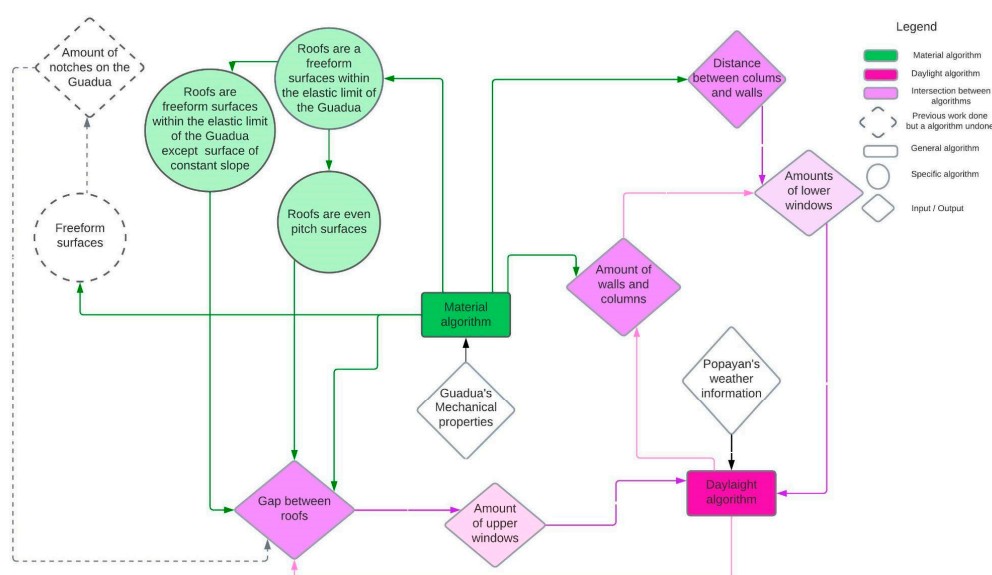

**Figure 5.** Specific diagram of numerical answer interactions between algorithms.

## 3. Results

The results for the geometric criteria will be presented first, followed by the material criteria and then the daylight criteria. They are described from the simplest to the most-complex criteria.

### 3.1. Geometric Criteria

We will first go over the history of the spiral and why certain spiral variants were chosen. Also, we will show the geometric methodology used for the generation of a constant-slope surface. In the second and third section, we will describe how the site's location on a steep slope affects the volume, and how ADA standards are complied. In the fourth section, we will show how the geometry is affected by the location of the fireplace.

#### 3.1.1. A Geometry Based on a Spiral

Several sketches were made initially that did not consider the spiral as the initial morphology. The design was then refined, with the help of the panel of experts. The original concept called for a volume with the spiral's geometry only in the plan. Later, some ideas were developed where the roof's slope increases toward the spiral's center. This modification enables the community to see the spiral's geometry when seen far from the volume.

Various spiral-based geometries were created, based on many different variants of spirals, in the knowledge that the one drawn and described by the Misak community is not very accurate. Two of them are shown in the following figures. The first one is the Fibonacci spiral depicted in Figure 6a, which approximates to the golden spiral, using quarter-circle

arcs inscribed in squares derived from the Fibonacci sequence, and the second one is a freeform spiral, depicted in Figure 6b.

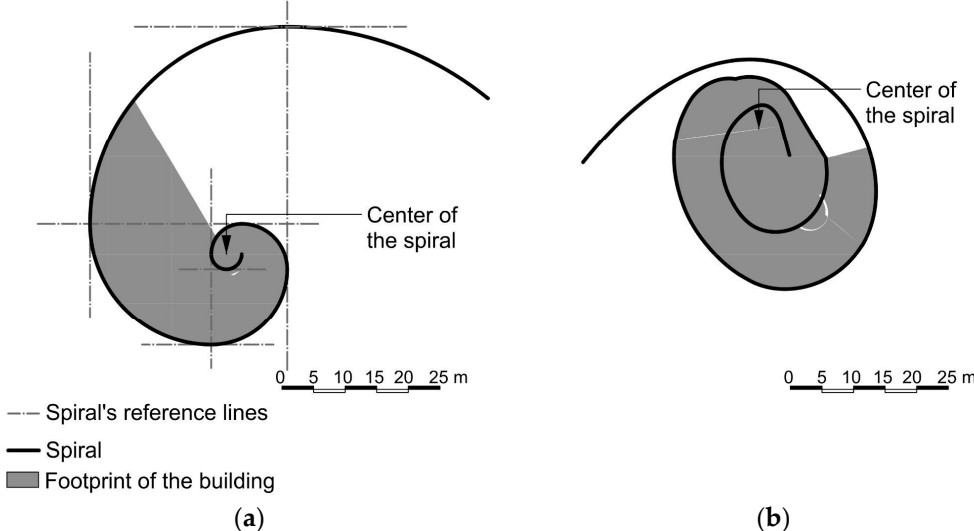

(**a**)                                            (**b**)

**Figure 6.** (**a**) Relationship between the Fibonacci spiral and the building in the plan. (**b**) Relationship between the freeform spiral and the building in the plan. Own elaboration.

Having as input any variant of spiral presented previously, an algorithm is constructed to build a freeform surface with bowed elements within the elastic limit of the Guadua (*Guadua angustifolia*), called "Variant A".

Having the same input, another algorithm, allowing the building of a surface of constant slope is constructed, called "Variant B". The geometric methodology for the generation of this surface starts from a closed planar boundary edge (the bank). This bank is split into two parts which are used to generate a tween/intermediate curve at an equal distance from both of them (a projection of the ridge on the reference plane). Next, a series of circles are drawn with their centers lying on the intermediate curve, and with them being tangent to the two parts of the bank. The center points are then connected by lines to their corresponding intersection points on the curves. The lines are subsequently rotated by a constant angle ($\theta$) about the tangents ($T$) at the intersection points and in the planes defined by the normals (N) and the binormals ($B$). The points at which the rotated lines intersect define the ridge of the surface. The surface of constant slope is generated through the bank, the ridge, and the straight lines that connect them, which is why it is also a ruled surface, as in Figure 7.

### 3.1.2. The Use of Terraces as an Adaptation to the Site

The chosen site has a steep slope, as described in Section 1. As a site adaptation, the algorithm was created to obtain a volume with multiple levels within the volume. In other words, this volume might be relocated to a different site with a different slope, and the algorithm will still adapt to the volume. In Figure 8a, there is shown a portion of the proposed volume, where there is a large area for activities like eating, dancing, and painting, as well as a small level area with the building's entrance.

### 3.1.3. Disabled Access ADA

The algorithm sets ramps with a maximum slope of 10%. This variable should be set to a different maximum slope which would be allowed if we applied the same algorithm in another country, and would have an impact on the geometry.

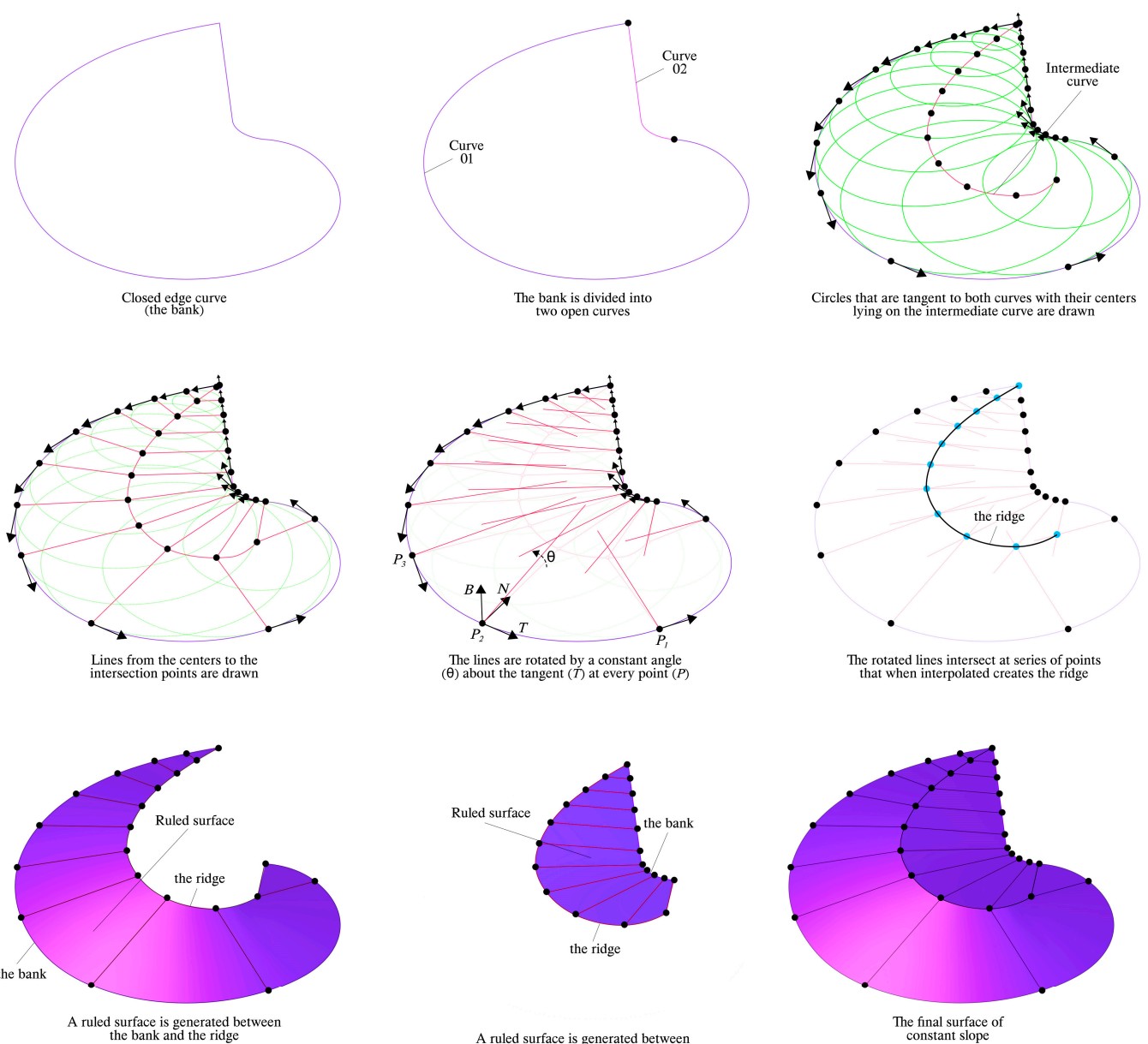

**Figure 7.** Construction methodology of a surface of constant slope. Own elaboration.

### 3.1.4. Fireplace Criteria

Because a fireplace that satisfies the needs of the Misak community must be included in the volume, as stated in Section 1. The architectural integration of the fireplace has been a crucial criterion for creating a comfortable and functional space within the building. The final algorithm was developed based on different criteria: thermal efficiency to avoid energy waste and ensure sufficient heating capacity, and good ventilation to ensure proper air intake and emission of smoke. The proposed volumes, with the fireplace at the center of the spiral, are depicted in Figure 9a,b. The point where the fire and ashes exit the building is located at the highest part of the roof. The proposed geometry not only provides a warm and comfortable atmosphere, but also becomes a significant design element, contributing to the overall aesthetics of the structure while ensuring the safety of the occupants.

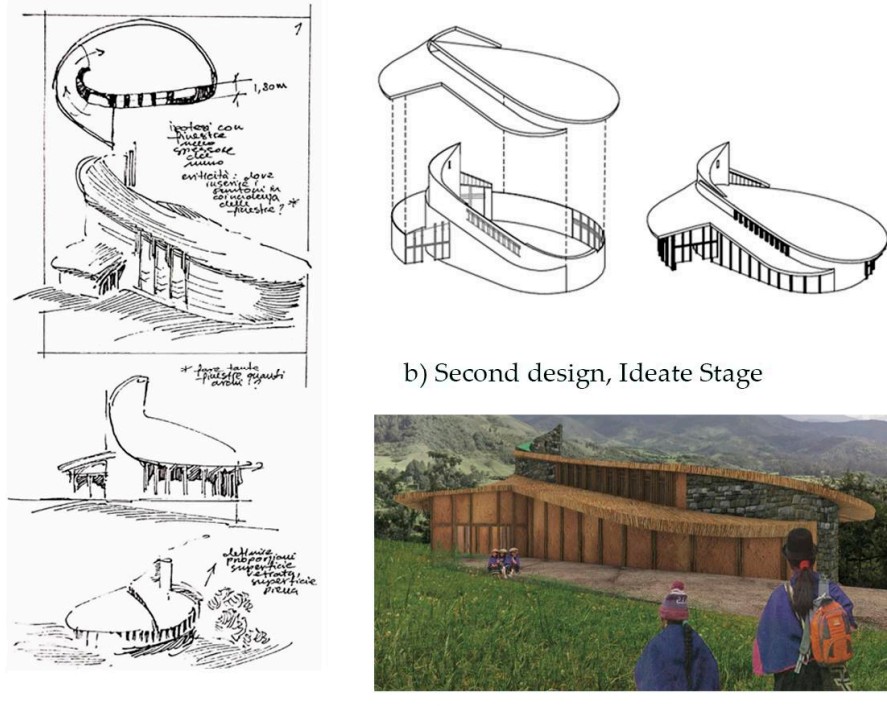

b) Second design, Ideate Stage

a) Initial design, Define Stage        c) Last design, Prototype Stage

**Figure 8.** "Design thinking" methodology. (**a**) First sketches of the proposed volume, after community interview. (**b**) Second drawings, after defining the first criterion. (**c**) Rendering, after algorithmic parametric design. Own elaboration.

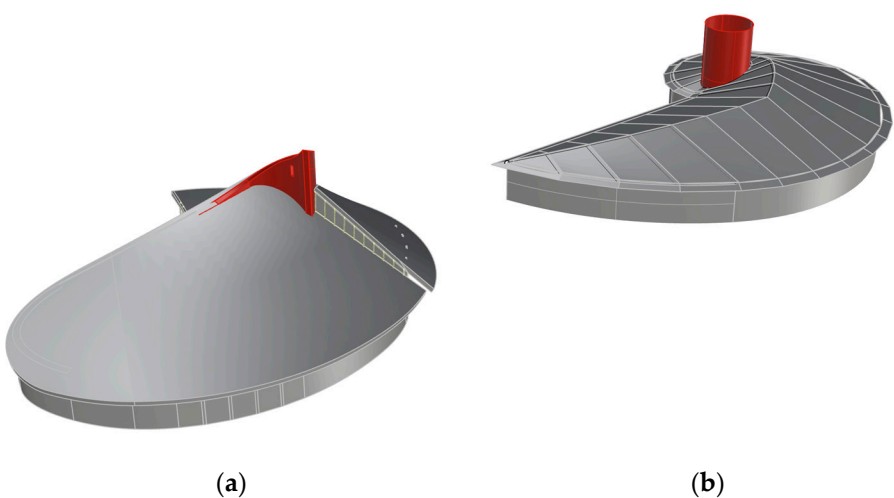

(**a**)                                    (**b**)

**Figure 9.** (**a**) Fireplace location of the building with a roof with a freeform surface. (**b**) Fireplace location of the building with a roof with an even-pitch surface. Own elaboration.

The possibilities are vast when geometric criteria are used. A group of experts has examined many of these options, and decisions have been made based on a general agreement.

### 3.2. Material-Informed Criteria

To explore the relationship between form, strength, and material, the morphological research seeks to integrate material characteristics. The material then assumes a crucial role in the design process of the form, becoming the driving force behind it.

### 3.2.1. Material within the Elastic Limit

The symbol of the spiral has an impact on the morphology and geometry of the building. The minimum radius of curvature of the Guadua (*Guadua angustifolia*) will be used as a parameter to achieve the targeted geometry, without using treatments involving changes in temperature or pressure. The material used, Guadua (*Guadua angustifolia*), in its natural form, has a limited elastic deformation capacity. The first structural criterion we take into account concerns the material shaping phase, i.e., the capacity of the Guadua rods to withstand large elastic deformations so as to form a grid with large curvatures.

The structure is shaped by bending. This causes the material to undergo significant deformation, which must remain within the elastic range. It is therefore necessary to check that the behavior of Guadua rods remains linearly elastic throughout the manufacturing phase.

In the case of the bending material, we consider two cases: the first involves shaping the material in its natural state—Variant "A"—the second involves using a bending technique to reduce the cross-sectional area to create curved structural elements.

For a given rod diameter $d$ we seek to obtain the smallest bending radius $r$. Considering a beam of Young's modulus $E$, the maximum bending stresses $\sigma^{max}$ are given by:

$$\sigma^{max} = E\frac{r}{R},$$ (1)

where $E$ is the Young's modulus of the material in MPa, $R$ is the radius of curvature, and $r$ is the outer radius of the Guadua.

By introducing the maximum elastic stress $\sigma_e$, we obtain the minimum radius of curvature as follows:

$$r^{min} = \frac{E}{\sigma_e R},$$ (2)

To calculate the bending moment to be applied to achieve the required curvature, we calculate the moment of inertia of a hollow tube as follows:

$$I = \pi\left(r^4 - (r - t)^4\right),$$ (3)

where $t$ is the outer radius of the Guadua.

The deflection to be applied is then equal to:

$$M = \frac{E\,I}{R},$$ (4)

Based on these formulae, the minimum radius of curvature is 9 mm and the bending moment to be applied is then 5.3 KNm.

Existing techniques that reduce the cross-sectional area to allow the material to be bent include the following:

1.  Rup-rup: this technique involves making V-shaped cuts (perpendicular to the fibers) along the entire length of the bamboo stalks.
2.  Split stalks: bamboo stalks are split (along the fibers) into wide, flat, rectangular slats.
3.  Lidi bundles: Bamboo stalks are cut (along the fibers) into thin cylindrical stalks. These techniques differ in terms of the final degree of curvature that the stalks can achieve.

In this research we are considering the rup-up technique. Whenever the minimum radius of curvature allowed by the material is exceeded, we will make notches. To quantify the reduction in material stiffness caused by this technique, which modifies the inertia of the cross-section, we calculate the length of the sides for a notch angle $\alpha$ is 6° and a circle radius $R$ is 20.5 m.

To identify the distance between notches and their angle, we consider a polygonal line made up of segments of equal length forming equal angles; as soon as such a line is made up of more than one segment, there is a single circle that passes through all the vertices of the line, as shown in Figure 10.

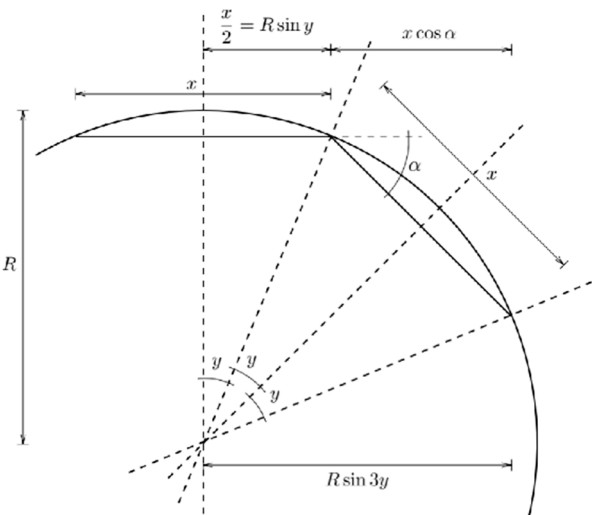

**Figure 10.** Method to identify the distance between notches. Own elaboration.

The following system of equations gives us the step:

$$\frac{x}{2} = R \sin \sin y, \tag{5}$$

$$\frac{x}{2} + x \, \mathrm{Cos}[\alpha] = R \sin \sin 3y, \tag{6}$$

The step $p$ will then be equal to 1.8 m.

We use the same formula to check the radius of the circle obtained with the given spacing $p$ and the angle of the notches given by the following system of equations, as shown in Figure 11:

$$\frac{p}{2} = x \sin y, \tag{7}$$

$$\frac{p}{2} + p \, \mathrm{Cos}[\alpha] = x \sin 3y, \tag{8}$$

where $x$ is the radius of the circle circumscribing a polygon given the angle between sides $\alpha$ and their length $p$, and $2y$ is the angle in this circle corresponding to the chord of length equal to the side of the polygon.

To calculate the maximum stress generated by the curvature with the notches, we also need to know the depth of the notches, where $h$ is equal to 80 mm.

Finally, we calculate the elongation of the outer fibers on the assumption that they follow arcs of a circle whose center is the apex of the notch, and remaining straight everywhere else in the interval $p$.

$$\varepsilon = \frac{\alpha(2r - h)}{p}, \tag{9}$$

The stress on the same fibers will be equal to:

$$\sigma = E \, \varepsilon, \tag{10}$$

The maximum stress calculated is 10.5 MPa, so it can still be positioned into the range of maximal values of material (14 MPa $< \sigma <$ 18 MPa).

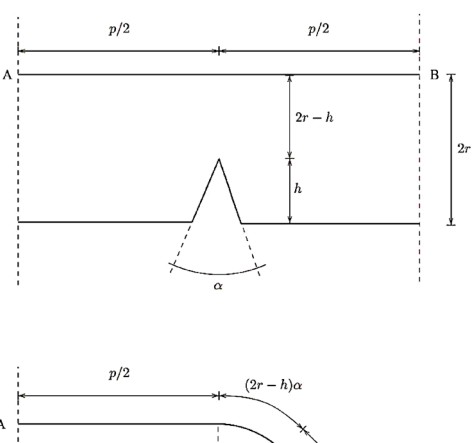

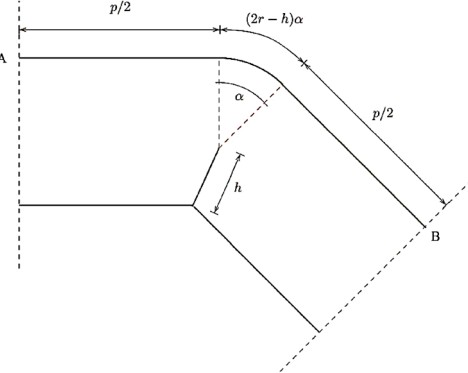

**Figure 11.** Method for the spacing between notches (*p*) and angle of notches given (*α*). Own elaboration.

### 3.2.2. Dimensioning Structural Elements of Constant-Slope Surfaces

To explain the methodology of the structural analysis, we show an iteration in the domain of the following surface of constant slope, "Variant B", shown in Figure 12. The design of the structure is based on 20 planar trusses, known as "English trusses ", according to Nicholson. The various elements of the structure are made up of a single Guadua (*Guadua angustifolia*) rod.

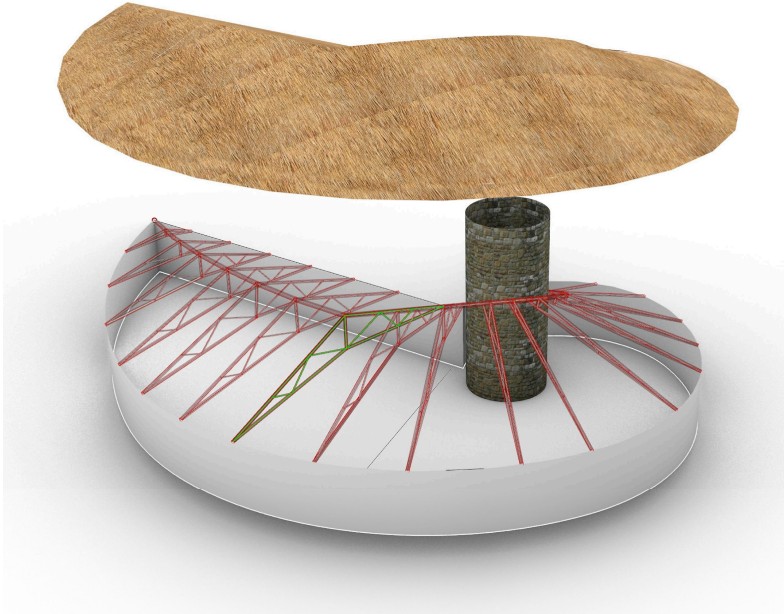

**Figure 12.** Building with a surface of constant slope, or "Variant B".

The structure with the longest span is studied to assess its behavior under loading in the working limit state and in the ultimate limit state, as shown in Figure 13.

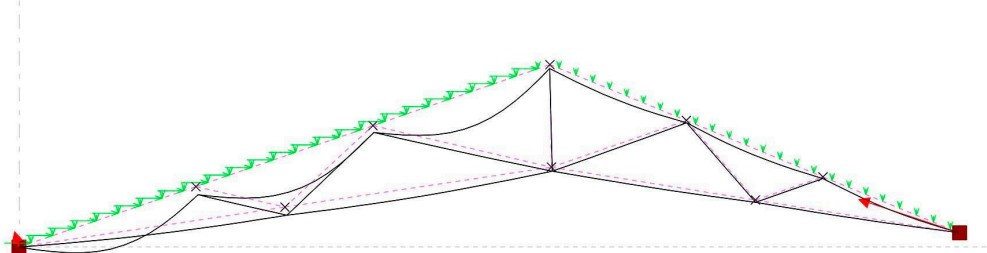

**Figure 13.** Displacements along the *x* and *y* axes of the critical beam of "Variant B". Own elaboration.

The main external loads to be considered for the project are the self-weight of the structure, the weight of the roof covering, and the variable loads (snow and wind). We consider the self-weight of the structure of Guadua rods and for each assembly (5 N). The weight of the roof covering is chosen as a flat rate of 140 N/m$^2$, which is an order of magnitude for the type of straw covering considered. The reference values for wind pressures are those of the project site (Silvia, Colombia), which are 600 N/m$^2$ for wind. The most unfavorable wind direction here is oriented from the north; this will be the only wind case studied. Snow loads do not apply in this case, because the area is not exposed to snowfall.

Based on these loads, we calculate the load *q* at the service limit state (*SLS*):

$$qSLS = pp + proof + pwind ,  \tag{11}$$

where *pp* is the load of the first floor, *proof* is the load of the roof, and *pwind* is load of wind.

The load *q* at the ultimate limit state (*USL*) considering the criterion of maximum stress. The loads are multiplied by safety factors (1.35 for permanent loads and 1.5 for variable loads).

$$qUSL = 1.35pp + 1.35proof + 1.5\ pwind,  \tag{12}$$

At the serviceability limit state, we verified the maximum displacement. and imposed the usual limitations for wooden structures or structures similar to bamboo. This way, the vertical displacements must be less than 1/240th of the span [24], and horizontal displacements must be less than 1/125th of the height.

If we assume that the average span of the structure is 18 m and its maximum height is 3.50 m, then the vertical displacements of the structure are acceptable if they remain below 7.5 cm, and the horizontal displacements should be below 3.0 cm. Figure 13 shows the displacements along the *x* and *y* axes. It can be observed that both horizontal and vertical displacements (*dx* = 0.03 cm and *dy* = 0.12 cm) remain within acceptable limits throughout the structure.

For safety, we set the requirement that the stress in the material must not exceed 50% of the ultimate strength (14 MPa for compression and 18 MPa for traction) [24].

We find that the maximum stress exceeds the failure stress.

To verify the structure in the ultimate limit state, we decided to double the rods for the bottom chords and the top chords. The results show that the maximum stress remains lower than that at failure, as shown in Figure 14.

### 3.3. Daylight Criteria

For the purposes of this study, two distinct algorithms were developed: (1) a building with a roof that has a freeform surface, named "Variant A" from now on, and that is shown in Figure 15a, and (2) a building called "Variant B", shown in Figure 15b. One result from each algorithm will be subjected to simulations of daylight.

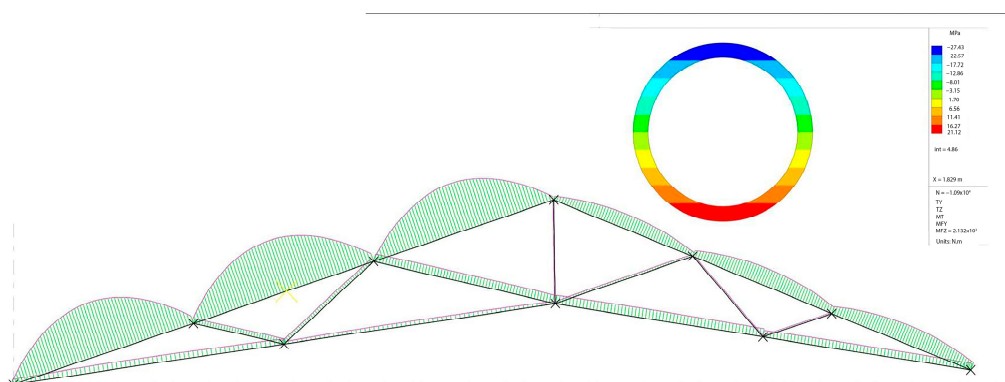

**Figure 14.** Maximum stresses in the most stressed section (top chord) of "Variant B". Own elabora-tion.

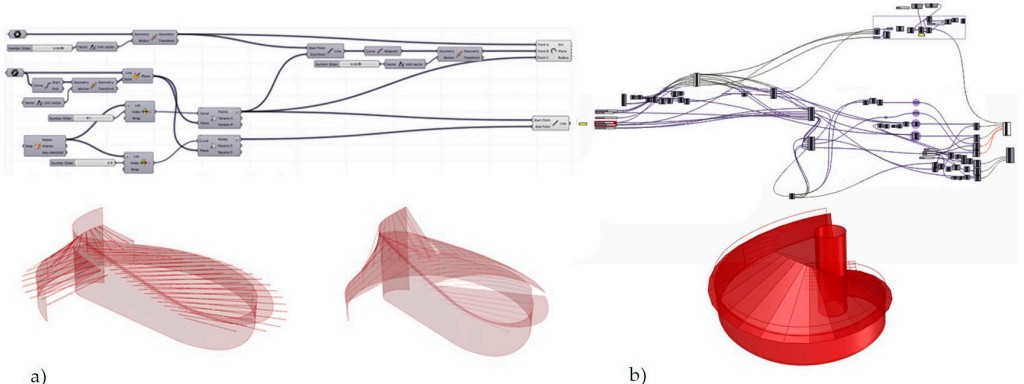

a)                                                                b)

**Figure 15.** (**a**) Parametric model and its visualization of "Variant A". Drawings by students of the course "Digital, optimal and smart structures", directed by authors of this paper. (**b**) Parametric model and its visualization of "Variant B". Own elaboration.

First, we compared two daylight simulations carried out under identical circumstances: the event took place on 21 June at 12 a.m. in Popayan, Colombia, which is the closest location to Silvia, Colombia. The software's closest weather information source is located here. Figure 16a shows a daylight simulation of the "Variant A" in the plan, with a few windows strategically located to take advantage of the best views, in addition to a gap between roofs. Figure 16b shows a daylight simulation of the "Variant B" in the plan, with the same number of windows, because it has no gap between roofs, since it is a single element. Furthermore, it should be noted that both volumes have a similar size in terms of plan and height. Due to the space between the roofs, we noticed that the image on the left not only has areas where the light ranges from 1000 to 2000 lux, but it also has a small spot of 2000 lux in the center of the area. We noticed that the image to the right has nearly the same amount of light. Finally, the distance between the roofs essentially has no bearing on the results.

Second, we conducted a comparison under the same circumstances, with the exception that the time was changed from 12 a.m. to 4 p.m. In Figure 17a, there is a daylight simulation of the "Variant A" in the plan. In Figure 17b is shown a daylight simulation of the "Variant B" in the plan. We see that the area of light in the image on the left is significantly larger, ranging from 1000 to 2000 lux, whereas the area of light in the image on the right is only about one-third as large. This occurs because of the geometry of the roof and the cantilever over the windows.

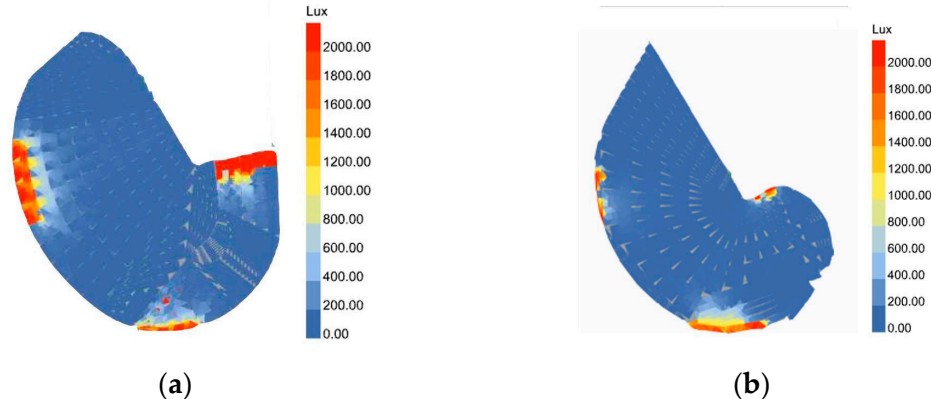

**Figure 16.** (**a**) Daylight ground floor plan simulation of "Variant A" at 12 a.m. (**b**) Daylight ground floor plan simulation of "Variant B" at 12 a.m. Own elaboration.

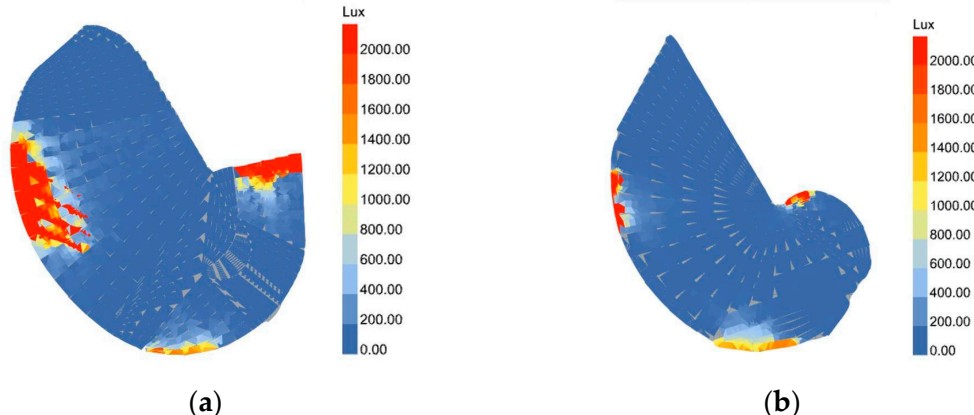

**Figure 17.** (**a**) Daylight ground floor plan simulation of "Variant A" at 4 p.m. (**b**) Daylight ground floor plan simulation of "Variant B" at 4 p.m. Own elaboration.

Third, we made a comparison for 4 p.m. on the same day every year. Both depict the same "Variant A". Figure 16a has the same strategic windows as in Figures 16b and 18 has windows all around the perimeter. We note that the area where daylight ranges from 1000 to 2000 lux goes almost up to the center of the space, making the left side the most logical place to place windows. The daylight, which ranges from 1000 to 2000 lux, only reaches a few meters into the rest of the building. In conclusion, the decision to only install a few strategically placed windows was correct because, for a low cost, the amount of light inside the building was adequate for the planned activities.

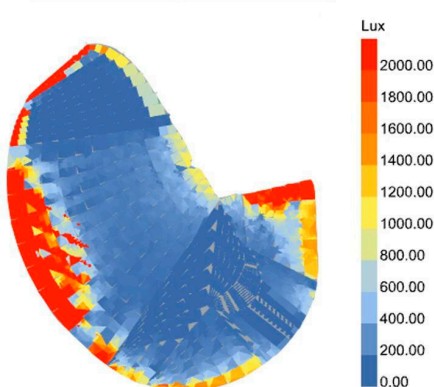

**Figure 18.** Daylight ground floor plan simulation of "Variant A", with windows all around the perimeter, at 4 p.m. Own elaboration.

To conclude, in terms of daylight, the best possibility is "Variant A" because it allows a better interior natural lighting. Somehow, even if a gap between roofs is necessary from a geometric point of view, this gap should be filled with a non-structural wall because the use of upper windows is not advised. In addition, only a few windows placed strategically are recommended.

An interior rendering that is a synthesis of all the requirements asked for by the Misak community, and the research team objectives, is shown in Figure 19. Even if it is a simulation of the reality, what could be perceived in this Figure is how, with only a few strategic windows, the daylight atmosphere can be produced. The main structural material of the building is the Guadua (*Guadua angustifolia*). Other materials considered are: (1) "bahareque ", which t is an ancestral technique that uses Guadua (*Guadua angustifolia*) and mud, shown on the right of the Figure 19, and (2) stone, shown on the left of the Figure 19, proving that the choice of secondary materials is also consistent with the environmental perspective.

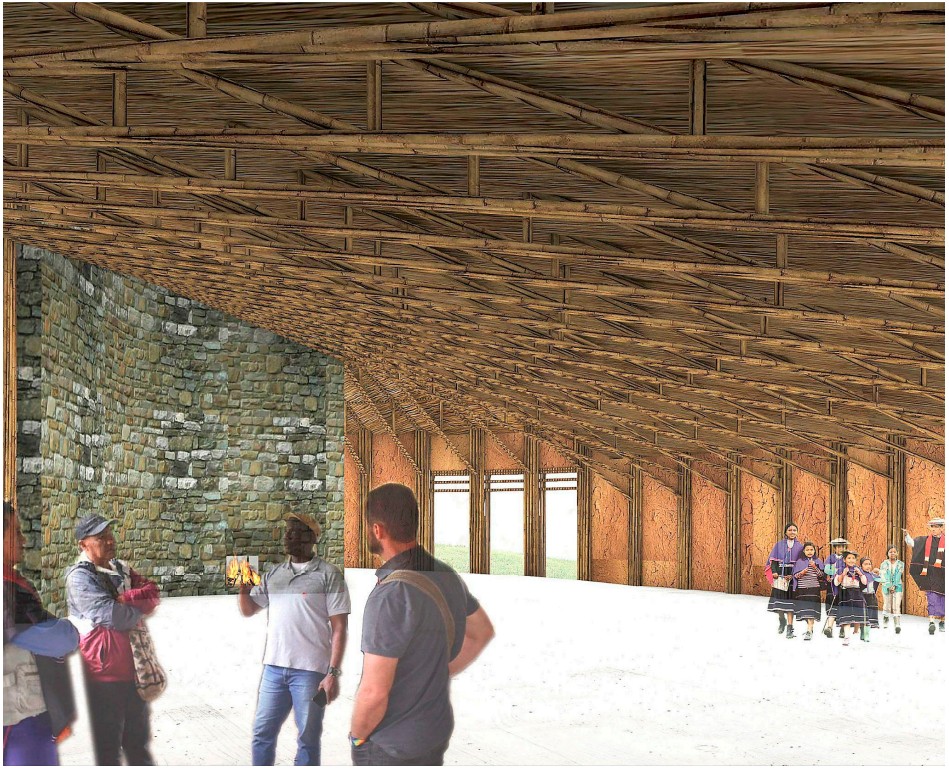

**Figure 19.** Interior rendering of volume with a roof with the geometry of a freeform surface with strategic windows. Drawings by students of the course "Digital, optimal and smart structures", directed by authors of this paper.

## 4. Discussion

Collaborative and participatory aspects of the methodology make this research innovative compared to other methodologies used in parametric design, where cultural and social aspects are usually set aside. The combination of variables obtained during the interviews with the Misak community in the first stage of the design-thinking methodology "Empathize", are categorized according to the cultural heritage theory of Kamelnia [16]. This makes it possible to match the parametric model in various situations where the cultural heritage is an integral component of the community's traditional architecture. Each category of cultural heritage corresponds to a particular algorithm, explained as follows: (1) "Cultural, historical, vernacular", corresponding to the geometric algorithm (case study: the fireplace); (2) "Form and symbol", corresponding to another part of the geometric algorithm (case study: the spiral); (3) "Architectural signatures," corresponding to the material algorithm (case

study: "bahareque", as the construction system used for walls); and (4) "Material, Colors", corresponding to another part of the material algorithm (case study: the Guadua (*Guadua angustifolia*). This classification simplifies how to apply the same parametric model—or one that only requires a minor adjustment—in a different context.

This research is an example of a "divergent" design, developing a framework associated with creativity in a similar way to the research presented by Suyoto [8]. In a similar way to Zhao and De Angelis [9], which creates hierarchical relationships between variables, the geometric algorithm is on the first level and the structural and daylight algorithms are on a second level. The last two algorithms are on the same level in the hierarchy.

In addition, it is similar to the Tabadkani [10] research, where the design process considered four stages: parameter-based modelling that was born from the Empathize stage from the community interviews; performance evaluation, which arises from the process of defining the criteria based on cultural heritage and material; simulations, which are based on the ideational stage, in which the first models are created, based on the algorithms; and a final optimization in the prototyping stage, in which models are created for rapid iterations. We developed the exact same stages in the same order in the material and daylight algorithms. Nevertheless, in the geometric algorithm, we carried out only the first stage.

The limit of the research is the algorithm when the roof does not meet the minimal radius of curvature and some notches in the Guadua (*Guadua angustofolia*) are needed. This part of the material algorithm has not been developed, even if all previous calculations have been made, because the Misak community lacks the skilled labor necessary to construct this type of construction. Nevertheless, it would have been interesting to analyze the results of this specific algorithm.

Future work on this research will focus on developing complementary algorithms related to (1) wind, (2) acoustics, (3) interior lighting, and (4) exterior lighting. This will allow us to combine information with the algorithms created for the purposes of this paper. If those algorithms were combined, the accuracy of the solution—in this case, the building's design—would be higher. The research goal in the future is also to optimize stone walls and "bahareque", which could be done in this Colombian region.

**Author Contributions:** Conceptualization, S.S.-V. methodology, S.S.-V., N.C.-L. and R.Z.; software, A.A.; validation, N.C.-L.; formal analysis, A.A.; structural design and analysis, R.Z.; daylight analysis, N.C.-L.; investigation, N.S.-G.; resources, N.C.-L.; writing—original draft preparation, S.S.-V.; writing—review and editing, N.C.-L.; visualization, A.A.; supervision, R.Z.; project administration, R.Z.; funding acquisition, N.C.-L. All authors have read and agreed to the published version of the manuscript.

**Funding:** This research was funded by the "Ministère de la Culture" in France. The APC of this research was funded by the ANID + InES Género+ INGE210029.

**Institutional Review Board Statement:** Not applicable.

**Informed Consent Statement:** Not applicable.

**Data Availability Statement:** Data are unavailable due to privacy restrictions.

**Acknowledgments:** We would especially like to thank Maurizio Brocato, who participated in this research (conceptualization, methodology and structural designs and analyses) but tragically passed away in January 2023. We also want to thank all the students who took the ENSA Paris-Malaquais course "Digital, optimal and smart structures", and the Misak community.

**Conflicts of Interest:** The authors declare no conflict of interest.

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
