# Peer review of "Parametric Integration of Multiple Criteria from a Cultural Heritage Perspective"

_applsci, doi:10.3390/app13169195_

Round 1

Reviewer 1 Report

in this research work, the authors present an approach of decision-making for multicriteria optimization. The purpose is to illustrate how parametric design may be a decision-making strategy for multi-criteria optimization.  We suggest to considerate the following comments to improve this works

1- the abstract should highlight the methodology, the main contribution and the principle outcomes,  authors are kindly invited to rewrite the abstract

2-The hypothesis is tested through the design of a Guadua pavilion for the Misak Community in Colombia. we thought that similar modelisation are existing. furthermore we think that this model is without any generalization ability. So we recommend to focus on generalization ability for the parametric model.

3-We are upset by the authors way in expressing their thoughts such this complex sentence : On one hand we present a methodology to find a geometry using a specific material, 164 not a methodology to find materials that are recommended for a group of geometries.

4-The figures should respect the alignment and professional presentation. Most of them seem like sketches and are not suitable for a scientific paper

5- the findings of the research should be more explained. Although he authors considerate several criteria, this work seems a simple modelisation project and need elaboration to fulfill  scientific project requirements

3- The citation form is not appropriate such in  “Tabadkani, 2019”

please update the citations

the paper requires sentences reformulation

Author Response

Response to Reviewer 1 Comments

Point 1: the abstract should highlight the methodology, the main contribution and the principal outcomes, authors are kindly invited to rewrite the abstract

Response 2: Thank you for pointing this out. We have rewritten the abstract including methodology, main contribution and principal outcomes.

Point 2: The hypothesis is tested through the design of a Guadua pavilion for the Misak Community inColombia. we thought that similar modelisation are existing. furthermore we think that this modelis without any generalization ability. So we recommend to focus on generalization ability for the parametric model.

Response 2: Agree. We have, accordingly, explain how the generalization ability for the parametric model relies on the possibility of matching it with the cultural heritage, been an integral component of the community's traditional architecture. This classification of the cultural heritage simplifies how to apply the same parametric model in a different context. 

Point 3: We are upset by the authors way in expressing their thoughts such this complex sentence : Onone hand we present a methodology to find a geometry using a specific material, 164 not amethodology to find materials that are recommended for a group of geometries.

Response 3: Thank you for pointing this out, we have eliminated this sentence.

Point 4: The figures should respect the alignment and professional presentation. Most of them seemlike sketches and are not suitable for a scientific paper

Response 4: Agree. We have; accordingly, we have eliminated almost all sketches. The only one that we left was include in Figure 8 with the purpose of illustrating the progression of the “design thiking” methodology. We also complement the paper with new image that we consider would help a better understanding.

Point 5: the findings of the research should be more explained. Although he authors considerateseveral criteria, this work seems a simple modelisation project and need elaboration to fulfill scientific project requirements

Response 5: Thank you for pointing this out, we have explained how each algorithm match a particular category of a cultural heritage classification proposed by the authors of on article we have referenced. In this way, it is no longer a case study, but parametric model has a generalization ability.

Point 6: The citation form is not appropriate such in “Tabadkani, 2019”

Response 6: Agree. We have, modified all citations with this format. We doublecheck how it should be done taking as reference a couple of articles from the Journal Applied Sciences.

Reviewer 2 Report

This paper is interesting and shows how to investigate different algorithms can interact with one another, and the hierarchy between them, a case study with pavilion design for the Misak community in Colombia was analyzed by using Guadua angustifolia a local bamboo material.

My detailed comments and recommendations related to the paper are presented below.

1. A graphical attempt is made in the paper to illustrate the methods used to interact between the algorithms,which is not detailed and clear enough.

2. Formulas used in the structural criteria section should be sourced according to the reference standard.

3. How to consider cultural heritage in digital design and parametrically design cultural heritage elements should be analyzed and elaborated in
this paper.

4. Bamboo as a biomass material, its material properties include physical properties, chemical properties, weathering resistance properties and susceptibility to insect damage, in this paper, the above characteristics are less analyzed.

5.Mismatch between the title of the paper and the content of the paper.As a case study, this paper does not support the topic as effectively as it should.In short, the paper title is too large compared to the case content.

Author Response

Response to Reviewer 2 Comments

Point 1: A graphical attempt is made in the paper to illustrate the methods used to interact between the algorithms, which is not detailed and clear enough.

Response 1: Thank you for pointing this out. The figure with the algorithm interactions was modified as you can see in the following image.  

Point 2: Formulas used in the structural criteria section should be sourced according to the reference standard.

Response 2: Agree. Accordingly, we modified formulas and their references to comply the reference standar.

Point 3: How to consider cultural heritage in digital design and parametrically design cultural heritage elements should be analized and elaborated in this paper.

Response 3: Thank you for pointing this out. With respect to how cultural inheritance was considered in the article, two indications were added. Firstly, the stages of the “design thinking” methodology were linked to the stages of parametric design. This allowed us to obtain the necessary information from the community regarding those characteristics most representative of their cultural heritage and a iteration process were conducted with the initial models. This information was organized based on a classification by Kamelnia and Hanachi where each cultural heritage category match an algorithm done with the purpose of this paper.

Point 4: Bamboo as a biomass material, its material properties include physical properties, chemical properties, weathering resistance properties and susceptibility to insect damage, in this paper, the above characteristics are less analyzed.

Response 4: Agree. Accordingly, Guadua physical properties, chemical properties, weathering resistance properties and susceptibility to insect damage were included compiling information from different references.

Point 5: Mistmatch between the title of the paper and the content of the paper. As a case study, this paper does not suppert the topic as effectively as it should. In short, the paper title is too large compared to the case content. 

Response 5: Thank you for pointing this out. The title was corrected as:

“Parametric integration of multiple criteria from a cultural heritage perspective”
